# Effect of Preoperative Dry Eye Treatment with Intense Pulsed Light with Meibomian Gland Expression on the Refractive Accuracy of Cataract Surgery in Patients with Meibomian Gland Dysfunction-Related Dry Eye: A Single-Center, Prospective, Open-Label Study

**DOI:** 10.3390/jcm14082805

**Published:** 2025-04-18

**Authors:** Tatsukata Kawagoe, Yuki Mizuki, Miki Akaishi, Masaki Takeuchi, Kazuro Yabuki, Seiichiro Hata, Akira Meguro, Nobuhisa Mizuki, Takeshi Teshigawara

**Affiliations:** 1Department of Ophthalmology, Yokohama City University School of Medicine, Yokohama 236-0004, Japan; kawagoe@yokohama-cu.ac.jp (T.K.); yujimizumizu0302@gmail.com (Y.M.); akaishi.mik.pz@yokohama-cu.ac.jp (M.A.); takeuchi@yokohama-cu.ac.jp (M.T.); akmeguro@yokohama-cu.ac.jp (A.M.); mizunobu@yokohama-cu.ac.jp (N.M.); 2Department of Ophthalmology, Saiseikai Yokohamashi Nanbu Hospital, Yokohama 234-0054, Japan; yabukazu@gmail.com; 3Department of Ophthalmology, Yokohama Sky Eye Clinic, Yokohama 220-0011, Japan; s.and.e.hata@gmail.com; 4Department of Ophthalmology, Tsurumi Chuoh Eye Clinic, Yokohama 230-0051, Japan; 5Department of Ophthalmology, Yokosuka Chuoh Eye Clinic, Yokosuka 238-0008, Japan

**Keywords:** cataract surgery, intra-ocular lens (IOL), postoperative refractive error, keratometry, meibomian gland dysfunction (MGD), dry eye, intense pulsed light (IPL), tear film stability, ocular surface conditions, refractive accuracy

## Abstract

**Objective:** This research seeks to investigate the effects of preoperative intense pulsed light with manual meibomian expression (IPL-MGX) on the refractive accuracy of cataract surgery on dry eyes with meibomian gland dysfunction (MGD-related dry eyes). **Methods:** Fifty-six MGD-related dry eye cases planned for cataract surgery were analyzed. IPL-MGX (four times at 2-week intervals) was performed before preoperative examination. Axial length (AL), anterior chamber depth (ACD), corneal curvature (mean-K), tear break-up time (TBUT), superficial punctate keratopathy in the central cornea (C-SPK), corneal higher-order aberrations (HOAs), and predicted postoperative spherical equivalent (P-SE) were evaluated pre- and post-IPL-MGX. The postoperative subjective spherical equivalent (S-SE) was evaluated after one month. The absolute difference between the S-SE and P-SE was considered an indication of P-SE accuracy. Changes in all the variables were assessed before and after IPL-MGX treatment. **Results:** No significant differences were observed in AL or ACD (*p* = 0.85, 0.56). The differences in mean-K, TBUT, C-SPK, and HOAs were significant (*p* < 0.01). P-SE accuracy based on post-IPL-MGX data was significantly higher than that based on pre-IPL-MGX data (*p* < 0.01). P-SE accuracy was within 0.25 diopters (D) in 14.3% of pre- and 55.4% of post-IPL-MGX, within 0.5D in 55.4% of pre- and 92.9% of post-IPL-MGX, within 0.75D in 98.2% of pre- and post-IPL-MGX, and within 1.0D in 98.2% of pre- and 100% of post-IPL-MGX. In the range of 0.25 and 0.5 D, the accuracy of P-SE was significantly higher in post-IPL-MGX (*p* < 0.01). **Conclusions:** Preoperative IPL-MGX considerably improved the predicted postoperative refraction accuracy in patients with MGD-related dry eye undergoing cataract surgery.

## 1. Introduction

Cataract surgery replaces patients’ crystal lenses with artificial intra-ocular lenses (IOL). Approximately 10% of patients are dissatisfied with the outcomes of cataract surgery [1,2]. Blurred vision is the most common cause of postoperative dissatisfaction (68%), with postoperative refractive error often being a major contributing factor [3,4,5].

Postoperative refractive outcomes depend on accurate IOL power calculations and preoperative biometric measurements such as axial length (AL), anterior chamber depth (ACD), and keratometry, which are the most crucial factors influencing postoperative refractive outcomes [6,7].

Owing to advancements in optical coherence tomography (OCT), the precision of AL and ACD measurements has improved. Nonetheless, the precision of keratometry measurements has not significantly improved [8]. Inaccurate keratometry measurements are mainly due to ocular surface irregularities and tear film instability, both of which are mainly caused by dry eyes [9,10,11], potentially inducing intra-patient discrepancies in keratometry measurements [12,13,14]. These conditions can also be caused by meibomian gland dysfunction (MGD)-related dry eyes [15].

A previous study reported a disparity of up to 0.5 D in the results of IOL power calculations among patients with dry eyes at different visits [16]. A preoperative error of 0.5 D in IOL power calculation can lead to a postoperative refractive error of as much as 1.17 D, which has a remarkable effect on postoperative visual outcome [17].

Concerning the epidemiology of patients with dry eyes scheduled for cataract surgery, the Prospective Health Assessment of Cataract Patients’ Ocular Surface (PHACO) study reported that rapid tear break-up time (TBUT) was observed in approximately 60% of patients planning cataract surgery, and central corneal staining was seen in approximately 50% of these patients [18]. MGD was defined as abnormal meibum composition and secretion [19]. Approximately 86% of patients with dry eyes have been reported to have concomitant MGD [19]. Other studies indicated that MGD-related dry eye prevails in approximately 75% of patients planning cataract surgery [20,21].

Intense pulsed light (IPL) is commonly used to treat skin conditions such as rosacea and acne [22]. Extensive research has demonstrated the usefulness of IPL for MGD-related dry eye [23,24,25]. Furthermore, the use of IPL or manual meibomian expression (MGX) alone is not as effective as a combination of these techniques [26,27].

Based on these reports, it is speculated that preoperative IPL-MGX treatment of MGD-related dry eye in patients preparing for cataract surgery can alleviate their tear film stability and ocular surface conditions, leading to improved postoperative refractive accuracy.

Therefore, the primary objective of this study was to investigate the efficacy of preoperative treatment for MGD-related dry eye using IPL-MGX in improving the refractive accuracy of cataract surgery.

## 2. Materials and Methods

### 2.1. Study Design and Patients

This prospective, open-label study conducted at a single center involved Japanese patients scheduled for cataract surgery who had been diagnosed with dry eye and MGD. The STROBE checklist for cross-sectional studies was used to write this research paper [28]. The following inclusion and exclusion criteria were used to select patients diagnosed with cataract. The inclusion criteria were as follows: (1) The diagnostic criteria for dry eye established in Japan, encompassing TBUT ≤ 5 s and dry eye symptoms such as eye discomfort and visual disturbance, were used to diagnose dry eye [29]. The Japanese version of the Ocular Surface Disease Index was used to assess dry eye symptoms [30]. In this study, dry eye symptoms were not quantified, as this was not a requirement. (2) Patients were also diagnosed with hyposecretory MGD using the Japanese MGD diagnostic criteria, which included the following three factors: findings indicating orifice blockage, abnormalities around the orifices, and the presence of symptoms [20].

Irregularities in the area surrounding the orifices were evaluated as abnormal if one of the following criteria was observed: anterior or posterior mucocutaneous junction replacement, lid margin irregularity, or vascular engorgement. If both meibomian gland orifice abnormalities and reduced meibomian secretion were observed, blockages were assessed as positive [20].

The exclusion criteria were as follows: (1) previous use of medications or punctual plugs before the study onset; (2) patients with skin, blepharitis, or dermatological problems; (3) history of contact lens use; (4) previous ocular surgery, trauma, or inflammation, corneal dystrophy or scarring, or any disorder that may lead to irregularities of the ocular surface; (5) patients with diabetes; (6) patients who opted for alternative preoperative dry eye treatments after enrollment.

### 2.2. MGD-Related Dry Eye Treatment

The allocation of IPL-MGX to either the right or left eye was determined using permuted block randomization.

IPL-MGX was performed four times preoperatively by the same ophthalmologist at 2-week intervals. During the treatment period, no type of dry eye medication was allowed. Possible adverse effects of IPL, such as redness, swelling, blistering, and scarring, were assessed at each treatment session by the ophthalmologist.

### 2.3. IPL-MGX Procedure

In all cases and procedures, the IPL device (M22 IPL; Lumenis Be, Yokne’am Illit, Israel) was operated in triple pulse sequence mode with the following settings: pulse duration of 6 ms, pulse delay of 50 ms, energy fluence ranging from 11 to 16 J/cm^2^, and a filter wavelength of 590 nm.

Patients were instructed to keep both eyes closed, and their eyes were protected using IPL-Aid eye shields (Honeywell Safety Products, Smithfield, RI, USA). After applying gel to the treatment area on the skin, the procedure was carried out in two steps.

Step 1: Twelve pulses were delivered using a double-pass technique to the infraorbital and lower eyelid area with a 15 × 35 mm guide light (Figure 1) [31].Step 2: Three pulses were applied using a single-pass technique to both the upper and lower eyelids with an 8 × 15 mm guide light (Figure 2) [31].

MGX was performed approximately 10 min after IPL treatment on both the upper and lower eyelids using meibomian gland forceps (Charmant, Fukui, Japan). To minimize discomfort during the procedure, a 0.4% oxybuprocaine hydrochloride ophthalmic solution was administered.

### 2.4. Examination of Ocular Surface Conditions, Biometric Parameters (AL, ACD, and Mean-K), and Accuracy of Predicted Postoperative Spherical Equivalent

TBUT was used to assess tear film stability [32,33], while superficial punctate keratopathy in the corneal center (C-SPK) and higher-order aberrations (HOAs) were used to assess ocular surface conditions [34,35,36,37]. The AL, ACD, and mean-K are considered the most important parameters for IOL power calculations [38,39]. The accuracy of the predicted postoperative spherical equivalent (P-SE) was evaluated by calculating the absolute difference between the subjective postoperative spherical equivalent (S-SE) and the P-SE, as measured using the IOLMaster 700 (Carl Zeiss Meditec AG, Jena, Germany). These variables were examined at baseline and approximately 1 week after the four IPL-MGX treatments. Surgery was performed approximately one week following preoperative examinations.

TBUT, HOAs, C-SPK, and the P-SE accuracy were compared before and after dry eye treatment with IPL-MGX to assess the impact of IPL-MGX on tear film stability, ocular surface conditions, and refractive accuracy.

TBUT and C-SPK were evaluated using a fluorescein dye. A fluorescein strip (Showa Yakuhin Kako Co., Tokyo, Japan) was moistened with saline, and any excess fluid was carefully removed prior to assessment. A strip was then placed on the inferior bulbar conjunctiva. The mean TBUT was calculated by measuring at two time intervals [40]. The extent of central corneal staining was assessed using the grading method established by the National Eye Institute/Industry Workshop [41]. C-SPK was scored on a scale from 0 to 3, as follows: 0 = none, 1 = mild, 2 = moderate, and 3 = severe. HOAs within a 4 mm corneal center were analyzed using CASIA 2 (Tomey Corporation, Nagoya, Japan), which offered a precise image of the optical status of the cornea using Zernike polynomials. HOAs were quantified as the total magnitude of third- to sixth-order aberrations and calculated using the root mean square method [42]. The AL, ACD, and mean-K were analyzed using an IOLMaster 700. Barrett Universal II was used to calculate the P-SE [43]. The same experienced and certified technician checked the subjective postoperative visual acuity 1 month postoperatively using the Early Treatment Diabetic Retinopathy Study Chart. The absolute difference between the P-SE and S-SE was considered the value of the accuracy of the P-SE (i.e., postoperative refractive accuracy).

### 2.5. Statistical Analysis

SPSS software (v. 29.0; IBM SPSS Statistics, Armonk, NY, USA) was used for all statistical analyses. A *p*-value of less than 0.05 was considered indicative of statistical significance. All variables were compared before and after IPL-MGX treatment. The normality of variable distributions was assessed using the Shapiro–Wilk test, which showed a non-normal distribution for all variables, apart from ACD. Additional statistical analyses were performed based on the normality test results.

The Wilcoxon signed-rank test was employed to compare TBUT, C-SPK, HOAs, AL, mean-K, and P-SE accuracy before and after dry eye treatment with IPL-MGX [44]. The ACD was compared using a paired-t test [45]. P-SE accuracy in both the pre- and post-dry eye treatment was broken down into four categories: within 0.25 diopters (D), within 0.5D, within 0.75D, and within 1.0D. The relative proportions of each category were compared using a chi-square test [46].

G*Power (v. 3.1.9.7) was used to perform a post hoc analysis to assess the power of sample size. The post hoc power analysis indicated that the sample size (n = 56) provided 99.9% statistical power to detect a large effect size (Cohen’s d = 0.86) at a significance level of 0.05.

## 3. Results

### 3.1. Baseline Patient Characteristics

A total of 56 eyes of 56 subjects (26 male and 30 female) with MGD-related dry eye were selected for IPL-MGX treatment prior to preoperative examination for cataract surgery. The mean age was 74.4 ± 6.3 (range, 59–85). The nurses ensured that all patients followed all regiments in this study. Possible adverse effects of IPL, such as redness, swelling, blistering, and scarring, were not observed in any patient throughout the study.

### 3.2. Biometric Variables Pre- and Post-IPL-MGX Treatment

Table 1 shows the relevant biometric variables for the IOL power calculation pre- and post-IPL-MGX treatment. No significant difference was observed in the AL and ACD comparing the pre-IPL-MGX and post-IPL-MGX treatments (*p* = 0.86 and 0.56, respectively) (Table 1). However, a significant difference was observed in the mean-K between the pre-IPL-MGX and post-IPL-MGX treatments (*p* < 0.01) (Table 1).

### 3.3. Tear Film Stability and Ocular Surface Conditions Pre- and Post-IPL-MGX Treatment

Table 2 shows variables indicating tear film stability and ocular surface conditions in pre-and post-IPL-MGX treatments. Regarding tear film stability, the TBUT in post-IPL-MGX was significantly longer than in the pre-IPL-MGX treatment (*p* < 0.01) (Table 2). Concerning ocular surface conditions, significant improvements in both C-SPK and HOAs were observed following IPL-MGX treatment (both *p* < 0.01) (Table 2).

### 3.4. Postoperative Refraction Accuracy Based on Pre- and Post-IPL-MGX Treatment Data

Table 3 shows P-SE accuracy based on pre- and post-IPL-MGX treatment data. P-SE accuracy post-IPL-MGX treatment was significantly improved compared to that of pre-IPL-MGX treatment (*p* < 0.01) (Table 3).

### 3.5. Accuracy Within 0.25D, 0.50D, 0.75D, and 1.00D of Postoperative Refraction Based on Pre- and Post-IPL-MGX Treatment Data

Figure 3 shows P-SE accuracy within 0.25D, 0.50D, 0.75D, and 1.00D based on pre- and post-IPL-MGX treatment data. P-SE accuracy in each category was as follows: within 0.25D, 14.3% based on pre-treatment data and 55.4% based on post-treatment data; within 0.5D, 55.4% based on pre-treatment data and 92.9% based on post-treatment data; within 0.75D, 98.2% based on both pre- and post-treatment data; and within 1.0 D, 98.2% based on pre-treatment data and 100% based on post-treatment data (Figure 3).

P-SE accuracies within 0.25D and 0.50D based on the post-IPL-MGX treatment were significantly higher than those based on the pre-IPL-MGX treatment (both *p* < 0.01) (Figure 3).

## 4. Discussion

To the best of our knowledge, no previous study has revealed that preoperative IPL-MGX treatment of MGD-related dry eye improves postoperative refractive accuracy in cataract surgery. The current study found a remarkable improvement in P-SE accuracy after preoperative MGD-related dry eye treatment with IPL-MGX.

Previous studies indicated comparable results with eye drop dry eye treatments, such as a 5% lifitegrast and rebamipide ophthalmic solution [47,48]. However, the patients in these studies were diagnosed with dry eye but not MGD-related dry eye. Therefore, the dry eyes were merely treated with the eye drops. In contrast, all patients in the present study were diagnosed with both dry eye and MGD and were treated with IPL-MGX.

The most common types of MGD are obstructive and hyposecretory. A characteristic of these meibomian gland derangements is blockage at the duct ending [49]. In MGD, changes in tear film lipid composition can occur due to anatomical degeneration and pathophysiological changes, which can accelerate evaporative dry eye [50]. Epidemiologically, approximately 75% of patients planning cataract surgery have MGD-related dry eyes [20,21].

Multiple studies reported the negative impact of MGD-related dry eye on preoperative biometric measurements in cataract surgery [51,52,53]. Covita et al. demonstrated that more severe MGD can cause a shorter TBUT [54], indicating tear film instability [32,33]. Previous studies have concluded that tear film instability induces the tear film to break up quickly and thicken irregularly [55], which can increase HOAs and C-SPK in dry eyes [37,56,57], indicating poor ocular surface conditions [34,37]. These poor ocular surface conditions negatively affect the accuracy of preoperative corneal curvature measurements [58].

Regarding modern physical treatment for MGD-related dry eye, LipiFlow thermal pulsation, IPL, and Tixel were established by Lane et al. [59], Toyos et al. [23], and Safir et al. [60], respectively.

The efficacy of LipiFlow thermal pulsation and IPL on the improvement of tear film stability and ocular surface conditions has been reported [61,62], but not for Tixel. Furthermore, improvements in the accuracy of corneal curvature measurements with preoperative treatment for MGD-related dry eye using LipiFlow have been reported [15,52]. However, no previous findings have been reported regarding the improvement in postoperative refractive accuracy with preoperative MGD-related dry eye. Therefore, this is the first study to demonstrate that preoperative treatment for MGD-related dry eye considerably improves tear film stability and ocular surface conditions, leading to improved postoperative P-SE accuracy (i.e., refractive accuracy). In addition, no previous studies have reported the effects of preoperative IPL-MGX treatment on preoperative corneal curvature measurements. Therefore, this is the first study to show that preoperative treatment of MGD-related dry eye using IPL-MGX can increase the accuracy of preoperative corneal curvature measurements.

Given the fact that blurred vision is the most common cause of postoperative dissatisfaction (68%) and that postoperative refractive error is the main factor leading to blurred vision [3,4,5], preoperative treatment of MGD-related dry eye using IPL-MGX can be one of the standard preoperative procedures to raise postoperative patient satisfaction.

The combined application of IPL-MGX has been reported to be more effective than the use of either IPL or MGX alone [26,27]. Therefore, in the present study, the IPL-MGX combination therapy was administered. However, the exact mechanisms underlying the effectiveness of IPL therapy remain unclear. Nevertheless, the results are promising. Accordingly, several previous studies have speculated the following possible mechanisms [63,64,65]. The target tissue absorbs IPL flash energy, leading to a localized temperature increase. As the localized temperature increases, lipids are liquefied, leading to improved secretion and excretion. However, the localized temperature increase is not sufficient to liquify other substances such as keratin, which blocks the meibomian gland. Therefore, physical removal using MGX is required. For this reason, IPL and MGX complement each other. Additionally, the absorption of light energy by blood cells can result in the disruption of telangiectatic vessels along the lid margin. Blocking dilated capillaries and decreasing the production of inflammatory mediators can be detrimental to natural inflammatory processes in acinar cells. These processes are responsible for reducing edema. Furthermore, the bacterial load can diminish, and the meibomian gland structure can improve [63,64,65].

This study has several limitations. All participants were Japanese and aged between 59 and 85 years, which may restrict the generalizability of the results to other ethnic groups and age ranges. Moreover, the results may have been influenced by different medical staff and facilities. The Japanese MGD diagnostic criteria were used in the present study because they were designed such that MGD could be easily diagnosed without special equipment or knowledge, even by non-specialists [27]. The Japanese MGD diagnostic criteria did not include functional assessment, meiboscore, or morphological evaluation; therefore, these variables were not assessed in this study. Future studies may benefit from analyzing the relationship between postoperative visual outcomes and these assessments. The Japanese version of the dry eye criteria was used to diagnose dry eye, which is different from the Dry Eye Workshop version of the dry eye criteria [66]. This is mainly because, in comparison with other types of dry eye, the short tear break-up time (TBUT) subtype of dry eye is more prevalent in Asian populations [67,68]. This motivated the Asia Dry Eye Society to create new dry eye diagnostic criteria for Asian populations [69]. Ocular surface damage does not always accurately correlate with the extent of dry eye symptoms in short TBUT-type dry eye [70].

Therefore, additional research using other diagnostic criteria for dry eye is needed to evaluate whether these other criteria affect the findings.

In the current study, the same can be said regarding the assessment of C-SPK, which was conducted using a four-point grading scale established by the National Eye Institute/Industry Workshop. Other assessment methods for C-SPK, such as the Oxford Scheme [63], could also affect the findings. Moreover, the efficacy of alternative physical treatments, including IPL alone, MGX alone, and LipiFlow, for preoperative MGD-related dry eye treatment has been reported [45,46]. However, their efficacy on postoperative visual outcomes has not yet been analyzed. Therefore, a comparison of the efficacy of alternative physical treatment methods for postoperative visual outcomes would be interesting.

Finally, preoperative dry eye treatment using LipiFlow thermal pulsation has been shown to effectively alleviate tear film instability and ocular surface damage, not only in MGD-related dry eye cases but also in dry eye patients without baseline MGD [45]. These results indicate that IPL-MGX may also improve visual function in dry eyes without baseline MGD, which can serve as the basis for further research.

## 5. Conclusions

The present study illustrates that preoperative MGD-related dry eye treatment with IPL-MGX can be effective in improving tear film stability and ocular surface conditions, as demonstrated by the extension of TBUT and the decrease in HOAs and C-SPK scores, resulting in an improvement in preoperative corneal curvature measurement accuracy. These effects can increase postoperative refractive accuracy. The results of the present study suggest that other types of modern physical MGD-related dry eye treatments may also increase postoperative refractive accuracy in cataract surgery. Therefore, preoperative MGD-related dry eye treatment presents a promising approach to enhancing postoperative patient satisfaction, warranting further investigation and quantifiable evaluation in future research.

## Figures and Tables

**Figure 1 jcm-14-02805-f001:**
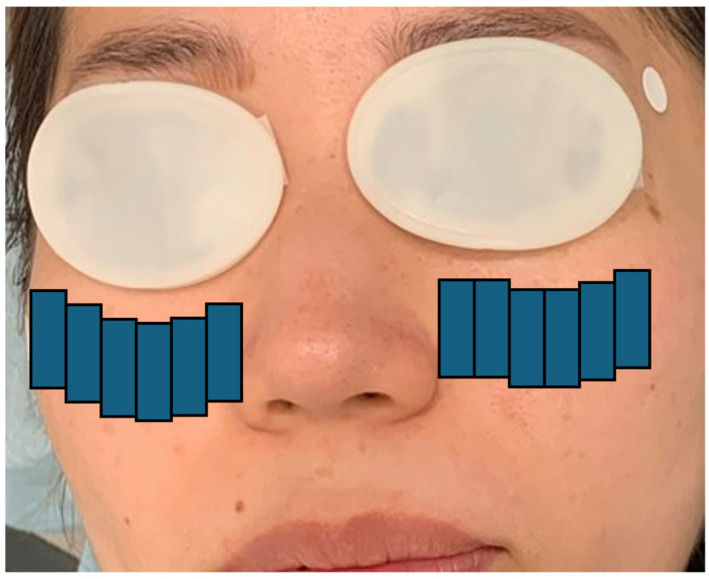
IPL protocol (Step 1): A double-pass technique involving 12 pulses was applied to the infraorbital and lower eyelid region using a 15 × 35 mm guide light. IPL refers to intense pulsed light. The diagram illustrates the procedure with eye patches; it is intended for illustrative purposes only and does not depict an actual patient [31].

**Figure 2 jcm-14-02805-f002:**
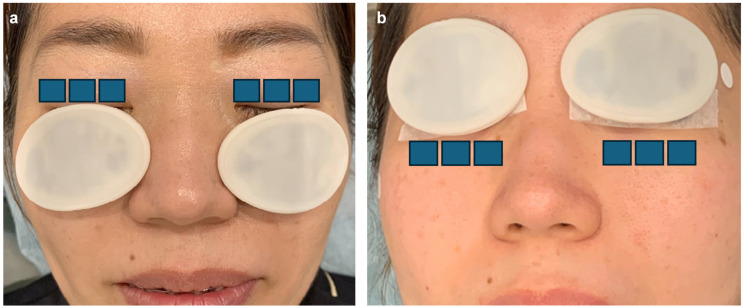
IPL protocol (Step 2): A single-pass technique consisting of three pulses was administered to the upper (**a**) and lower (**b**) eyelids using an 8 × 15 mm guide light. IPL stands for intense pulsed light. The diagram illustrates the procedure with eye patches for demonstration purposes only and does not represent an actual patient [31].

**Figure 3 jcm-14-02805-f003:**
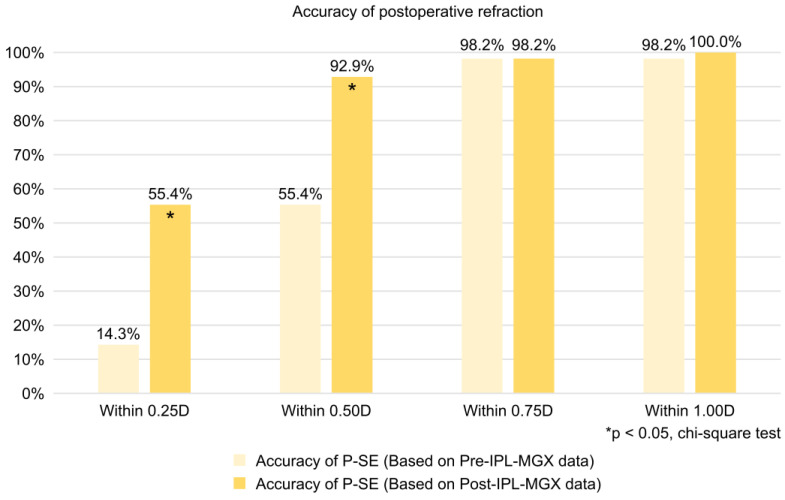
Accuracy of the predicted postoperative spherical equivalent (P-SE) evaluated using data obtained before and after IPL-MGX treatment. X-axis: the absolute difference between subjective postoperative spherical equivalent and predicted postoperative spherical equivalent. D, diopter. Y-axis: proportion of patients.

**Table 1 jcm-14-02805-t001:** Differences in AL, ACD, and mean-K.

	Pre-IPL-MGX Treatment			Post-IPL-MGX Treatment			*p*-Value
	Mean	SD	Median	Mean	SD	Median	
AL (mm)	23.86	1.49	23.38	23.84	1.46	23.38	0.85
ACD (mm)	2.98	0.36	3	2.99	0.36	3.02	0.56
Mean-K (D)	43.36	1.72	43.42	43.18	1.81	43.25	<0.01

AL, axial length; ACD, anterior chamber depth; IPL-MGX, combination treatment of intense pulsed light and meibomian gland expression; SD, standard deviation; Mean-K, corneal curvature.

**Table 2 jcm-14-02805-t002:** Differences in T-BUT, C-SPK, and HOAs between pre-and post-IPL-MGX treatment.

	Pre-IPL-MGX Treatment			Post-IPL-MGX Treatment			*p*-Value
	Mean	SD	Median	Mean	SD	Median	
TBUT (s)	2.64	1.05	3.00	4.18	1.16	4.00	<0.01
C-SPK	1.00	0.89	1.00	0.50	0.57	0.00	<0.01
HOAs (µm)	0.32	0.04	0.33	0.27	0.03	0.27	<0.01

TBUT, tear break-up time; C-SPK, superficial punctate keratopathy in the center part of cornea; HOAs, higher-order aberrations; IPL-MGX, combination treatment of intense pulsed light and meibomian gland expression; SD, standard deviation.

**Table 3 jcm-14-02805-t003:** Accuracy of P-SE.

	Pre-IPL-MGX Treatment			Post-IPL-MGX Treatment			*p*-Value
	Mean	SD	Median	Mean	SD	Median	
Accuracy of P-SE (D)	0.47	0.2	0.45	0.28	0.15	0.24	<0.01

P-SE, predicted postoperative spherical equivalent; S-SE, subjective postoperative spherical equivalent; accuracy of P-SE (D), the absolute difference between P-SE and S-SE; IPL-MGX, combination treatment of intense pulsed light and meibomian gland expression; SD, standard deviation.

## Data Availability

The raw data supporting the conclusions of this study will be made available by the authors upon request. The data are not publicly available due to ethical restrictions.

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
