# Peer review of "Effect of Preoperative Dry Eye Treatment with Intense Pulsed Light with Meibomian Gland Expression on the Refractive Accuracy of Cataract Surgery in Patients with Meibomian Gland Dysfunction-Related Dry Eye: A Single-Center, Prospective, Open-Label Study"

_jcm, 2025, doi:10.3390/jcm14082805_

Round 1
Reviewer 1 Report
Comments and Suggestions for Authors
This study evaluated the impact of preoperative intense pulsed light with manual meibomian expression (IPL-MGX) on refractive accuracy in cataract surgery for dry eye patients with meibomian gland dysfunction. Involving 56 cases, IPL-MGX improved measurements of corneal curvature, tear break-up time, superficial punctate keratopathy, and higher-order aberrations. Post-IPL-MGX, predicted postoperative spherical equivalent accuracy significantly increased, leading to better refractive outcomes for these patients. The important takeaway from the current manuscript is preoperative IPL-MGX treatment for MGD-related dry eye improves tear film stability and corneal conditions, enhancing refractive accuracy in cataract surgery and potentially increasing patient satisfaction with modern physical therapies.
- The data as shown in the manuscript is mostly observational and true mode of action for favorable ocular parameter changes due to IPL-MGX (pre-vs post-surgical procedure) is not clearly presented.
- The detailed demographic information for the 56 cases are missing. What are the proportion for sexes for the study cohort?
- Page 7, lines 275-279: The research manuscript needs to be data-driven and my recommendation would be to refrain from speculation if there is/are no direct supporting evidence confirming the speculation.
- The conclusion talks about post-operative patient satisfaction. What type of strategy or procedure (No data) was performed to ascertain or evaluate the patient satisfaction issue?
Minor:
- Figure 3 lacks Y axis legend description.
- Page 7, lines 273-275: Why is there a reference citation for speculating the mode of action for the IPL-MGX?
Author Response
Response to the Reviewer 1
Comment 1
The data as shown in the manuscript is mostly observational and true mode of action for favorable ocular parameter changes due to IPL-MGX (pre-vs post-surgical procedure) is not clearly presented.
Response
Thank you sincerely for your valuable comment. As you mentioned, the data in this manuscript are indeed mostly observational since the Japanese MGD diagnostic criteria did not include functional assessment, meiboscore, or morphological evaluation; therefore, these variables were not assessed in this study.
However, future studies may benefit from analyzing the relationship between postoperative visual outcomes and these assessments.
Nevertheless, the main value of this manuscript lies in its demonstration that preoperative IPL-MGX treatment for MGD-related dry eye significantly improves postoperative refractive accuracy in cataract surgery. This study primarily aimed to present evidence that IPL-MGX enhances tear film stability and ocular surface conditions, leading to more precise preoperative corneal curvature measurements and ultimately improving the accuracy of intraocular lens (IOL) power calculations.
Comment 2
The detailed demographic information for the 56 cases are missing. What are the proportion for sexes for the study cohort?
Response
Thank you sincerely for your comment. The proportion of sexes and ages for this study are described in lines 188-191 as follows:
“A total of 56 eyes of 56 subjects (26 male and 30 female) with MGD-related dry eye were selected for IPL-MGX treatment prior to preoperative examination for cataract surgery. The mean age was 74.4±6.3 (range, 59–85).”
Comment 3
Page 7, lines 275-279: The research manuscript needs to be data-driven and my recommendation would be to refrain from speculation if there is/are no direct supporting evidence confirming the speculation.
Response
Thank you for your helpful comment. This speculation has been mentioned in many previous studies. Therefore, we have revised this part as follows and have added two relevant references to the manuscript (lines 283-284).
“Accordingly, several previous studies have speculated the following possible mechanisms [63–65].”
We hope this revision addresses your concern.
Comment 4
The conclusion talks about post-operative patient satisfaction. What type of strategy or procedure (No data) was performed to ascertain or evaluate the patient satisfaction issue?
Response
Thank you for your pertinent comment. Given the fact that blurred vision is the most common cause of postoperative dissatisfaction
(68%), and postoperative refractive error is the main factor leading to blurred vision in cataract surgery [3–5], the results of this study indicate that preoperative MGD-related dry eye treatment with IPL-MGX can be one of the standard preoperative procedures to raise postoperative patients’ satisfaction.
However, since the improvement of the patient satisfaction was not investigated in this study, we have revised the relevant part in the Conclusion as follows:
“Therefore, preoperative MGD-related dry eye treatment presents a promising approach to enhancing postoperative patient satisfaction, warranting further investigation and quantifiable evaluation in future research.”
We hope this revision addresses your concern.
Minor
Comment 1
Figure 3 lacks Y axis legend description.
Response
Thank you sincerely for your meticulous review. However, a corresponding description of the Y-axis is indeed included in the figure legend as follows:
“X-axis: the absolute difference between subjective postoperative spherical equivalent and predicted postoperative spherical equivalent; D, diopter. Y-axis: proportion of patients.”
Comment 2
Page 7, lines 273-275: Why is there a reference citation for speculating the mode of action for the IPL-MGX?
Response
Thank you for your pertinent comment. As per our response to comment #3, numerous previous studies have proven the efficacy of IPL-MGX and speculated on the mechanism of IPL. Therefore, we have added two relevant references to this manuscript accordingly (Lines 283-284).
Ultimately, we would like to express our sincere gratitude to the editors and reviewers for their positive and constructive criticism. The manuscript has vastly benefited from your valuable and insightful comments and suggestions. We look forward to hearing from you and would be happy to address any further concerns, if required. We hope this further pushes the manuscript closer to publication in your esteemed journal.
Reviewer 2 Report
Comments and Suggestions for Authors
Dear Authors,
I congratulate you on your review in this interesting field. The topic is quite interesting because it emphasises the negative impact of dry eye condition caused by meibomian gland dysfunction on post-cataract surgery outcomes. Nevertheless, I suggest several aspects in order to enhance the manuscript.
- Line 73: The authors wrote the word "MGX". What is the meaning of this abbreviation? Abbreviations must first be written out in full before using the shortened form, although this may seem obvious.
- Concerning "MGD-Related Dry Eye Treatment" section: Were any adverse effects observed during the IPL application? If so, they should be described.
- Concerning "IPL-MGX Procedure" section: How much time elapsed between both steps? Was the same energy fluence used in both steps? What does the IPL treatment facilitate? Perhaps could it help open the meibomian gland?
-
How much time passed between the completion of the treatment and the post-IPL-MGX measurements before cataract surgery as well as these measurements and the surgery itself?
- Concerning the accuracy of P-SE: This aspect seems somewhat unclear. What do both parameters consist of, and how were they measured?
How was the accuracy P-SE determined before cataract surgery? Perhaps, did the authors consider subjetive pre-operative SE?
- In result section, the authors well-mentioned the number of eye taken into consideration. Nevertheless, how was the eye selection? Perhaps was it randomised, or was it based on the severity of the condition?
-
Why did the nurses ensure that all patients followed the treatment protocol? Perhaps was the treatment administered at the participants' home or was it carried out by an ophthalmologist at the clinic? What specific role did the nurses play in this process?
- Concerning tables presented in all the manuscript: The authors previously explained in detail the statistical tests in "Statistical Analysis" section, so adding a separate column reiterating the tests performed may not be necessary.
- The authors must review the cite number 68. Ginsburg, A.P. A new contrast sensitivity vision test chart. Am. J. Optom. Physiol. Opt. 1984, 61, 403–407. DOI:10.1097/00006324-513198406000-00011
Perhaps, it may be a confusion since this reference does not appear to be related with the topic of this manuscript.
Author Response
Response to Reviewer 2
Comment 1
Line 73: The authors wrote the word "MGX". What is the meaning of this abbreviation? Abbreviations must first be written out in full before using the shortened form, although this may seem obvious.
Response
Thank you sincerely for your meticulous review and pertinent comments. We would like to apologize for this oversight. Following your comment, we have added “manual meibomian expression (MGX)” in line 73 (new line number is 76).
Comment 2
Concerning "MGD-Related Dry Eye Treatment" section: Were any adverse effects observed during the IPL application? If so, they should be described.
Response
The manuscript indeed includes a corresponding statement on the adverse effects related to the treatment in lines 191-192 as follows:
“Possible adverse effects of IPL, such as redness, swelling, blistering, and scarring, were not observed in any patient throughout the study.”
We hope this response properly addresses your concern.
Comment 3
Concerning "IPL-MGX Procedure" section: How much time elapsed between both steps? Was the same energy fluence used in both steps? What does the IPL treatment facilitate? Perhaps could it help open the meibomian gland?
Response
Regarding the interval time between treatments, it was described in lines 113-114 and 126-127 as follows:
“IPL-MGX was performed four times preoperatively by the same ophthalmologist at 2-week intervals.”
“MGX was performed approximately 10 min after IPL on both the upper and lower eye-lids using meibomian gland forceps (Charmant, Fukui, Japan). “
Concerning the energy fluence of the IPL part of the procedure, the same mode was applied to all cases and procedures. Therefore, we have added “For all cases and procedures” to line 118 .
Additionally, the possible mechanism of IPL is described in lines 284-295 as follows:
“The target tissue absorbs IPL flash energy, leading to a localized temperature increase. As the localized temperature increases, lipids are liquefied, leading to improved secretion and excretion. However, the localized temperature increase is not sufficient to liquify other substances such as keratin, which blocks the meibomian gland. Therefore, physical removal using MGX is required. For this reason, IPL and MGX complement each other. Additionally, light energy absorption by blood cells can lead to the break-down of telangiectatic vessels in the lid margin. Blocking dilated capillaries and decreasing the production of inflammatory mediators can be detrimental to natural inflammatory processes in acinar cells. These processes are responsible for reducing edema. Furthermore, the bacterial load can diminish, and the meibomian gland structure can improve [63–65].”
Comment 4
How much time passed between the completion of the treatment and the post-IPL-MGX measurements before cataract surgery as well as these measurements and the surgery itself?
Response
Post IPL-MGX measurements were performed approximately one week after the completion of the treatment, which is stated in lines 149-150.
One week after the preoperative measurements, cataract surgery was performed. Therefore, we have added the following statement to lines 150-151. “Surgery was performed approximately one week following postoperative examinations.”
We hope this revision addresses your concern.
Comment 5
Concerning the accuracy of P-SE: This aspect seems somewhat unclear. What do both parameters consist of, and how were they measured?
How was the accuracy P-SE determined before cataract surgery? Perhaps, did the authors consider subjetive pre-operative SE?
Response
We have defined the accuracy of P-SE in lines 145-148 as follows: “The absolute difference between the subjective postoperative spherical equivalent (S-SE) and predicted postoperative spherical equivalent (P-SE) was used to determine the accuracy of P-SE.”
However, to further clarify P-SE, we have added “calculated using IOL Master 700 (Carl Zeiss Meditec AG, Jena,Germany)” in the relevant section as follows:
“The absolute difference between the subjective postoperative spherical equivalent (S-SE) and predicted postoperative spherical equivalent (P-SE), calculated using IOL Master 700 (Carl Zeiss Meditec AG, Jena,Germany), was used to determine the accuracy of P-SE.”
We hope this revision addresses your concern.
Comment 6
In result section, the authors well-mentioned the number of eye taken into consideration. Nevertheless, how was the eye selection? Perhaps was it randomised, or was it based on the severity of the condition?
Response
Thank you for your valuable comment. We have added the following text to lines 111-112 for further clarification:
“The allocation of IPL-MGX to either the right or left eye was determined using permuted block randomization.”
We hope this revision addresses your concern.
Comment 7
Why did the nurses ensure that all patients followed the treatment protocol? Perhaps was the treatment administered at the participants' home or was it carried out by an ophthalmologist at the clinic? What specific role did the nurses play in this process?
Response
Nurses were responsible for confirming if patients followed all regimens in this study. In response to your comment, we have revised lines 190-191 as follows:
“The nurses ensured that all patients followed all regimens in this study.”
We hope this revision is met with your satisfaction.
Comment 8
Concerning tables presented in all the manuscript: The authors previously explained in detail the statistical tests in "Statistical Analysis" section, so adding a separate column reiterating the tests performed may not be necessary.
Response
Thank you for your valuable suggestion. We acknowledge that the test types are already stated in the Statistical Analysis section and do not need to be repeated in the tables. Accordingly, we have removed the corresponding columns.
Comment 9
The authors must review the cite number 68. Ginsburg, A.P. A new contrast sensitivity vision test chart. Am. J. Optom. Physiol. Opt. 1984, 61, 403–407. DOI:10.1097/00006324-513198406000-00011
Perhaps, it may be a confusion since this reference does not appear to be related with the topic of this manuscript.
Response
Thank you sincerely for your insightful suggestion, which we completely agree with; this reference can come across as confusing. Therefore, we have replaced it with a more relevant reference as shown below.
“68. Kaido M, Kawashima M, Ishida R, Tsubota K. Severe symptoms of short tear break-up time dry eye are associated with accommodative microfluctuations. Clin Ophthalmol. 2017;11:861-869. Published 2017 May 5. DOI:10.2147/OPTH.S128939”
We hope this revision is met with your satisfaction.
Ultimately, we would like to express our sincere gratitude to the editors and reviewers for their positive and constructive criticism. The manuscript has vastly benefited from your valuable and insightful comments and suggestions. We look forward to hearing from you and would be happy to address any further concerns, if required. We hope this further pushes the manuscript closer to publication in your esteemed journal.
Round 2
Reviewer 1 Report
Comments and Suggestions for Authors
The authors have provided satisfactory responses to all my questions/comments. The revised manuscript has now been much improved.
Reviewer 2 Report
Comments and Suggestions for Authors
Dear authors,
The manuscript has been significantly improved.